# Dopamine and Serotonin Transporter Genes Regulation in Highly Sensitive Individuals during Stressful Conditions: A Focus on Genetics and Epigenetics

**DOI:** 10.3390/biomedicines12092149

**Published:** 2024-09-23

**Authors:** Fabio Bellia, Alessandro Piccinini, Eugenia Annunzi, Loreta Cannito, Francesca Lionetti, Bernardo Dell’Osso, Walter Adriani, Enrico Dainese, Alberto Di Domenico, Mariangela Pucci, Riccardo Palumbo, Claudio D’Addario

**Affiliations:** 1Department of Bioscience and Technology for Food, Agriculture and Environment, University of Teramo, 64100 Teramo, Italy; fabio.bellia@unich.it (F.B.); apiccinini1@unite.it (A.P.); eugenia.annunzi@unisi.it (E.A.); edainese@unite.it (E.D.); mpucci@unite.it (M.P.); 2Department of Innovative Technologies in Medicine and Dentistry, University “G. D’Annunzio” of Chieti-Pescara, 66100 Chieti, Italy; 3Center for Advanced Studies and Technology (CAST), University “G. D’Annunzio” of Chieti-Pescara, 66100 Chieti, Italy; loreta.cannito@unifg.it; 4Department of Social Sciences, University of Foggia, 71122 Foggia, Italy; 5Department of Brain and Behavioral Sciences, University of Pavia, 27100 Pavia, Italy; francesca.lionetti@unipv.it; 6Department of Biomedical and Clinical Sciences “Luigi Sacco”, University of Milan, 20019 Milan, Italy; bernardo.dellosso@unimi.it; 7“Aldo Ravelli” Center for Nanotechnology and Neurostimulation, University of Milan, 20122 Milan, Italy; 8Center for Behavioural Sciences and Mental Health, Istituto Superiore di Sanità, Viale Regina Elena, 299, 00161 Rome, Italy; walter.adriani@iss.it; 9Department of Psychological, Health and Territorial Sciences, University “G. D’Annunzio” of Chieti-Pescara, 66100 Chieti, Italy; adidomenico@unich.it; 10Department of Neuroscience, Imaging and Clinical Sciences, University “G.D’Annunzio” of Chieti-Pescara, 66100 Chieti, Italy; r.palumbo@unich.it; 11Department of Clinical Neuroscience, Karolinska Institute, 10316 Stockholm, Sweden

**Keywords:** highly sensitive person, stress, dopamine and serotonin transporter genes, epigenetics, biomarkers

## Abstract

**Background**: Coping with stress is essential for mental well-being and can be critical for highly sensitive individuals, characterized by a deeper perception and processing of stimuli. So far, the molecular bases characterizing high-sensitivity traits have not been completely investigated and gene × environment interactions might play a key role in making some people more susceptible than others. **Methods**: In this study, 104 young adult university students, subjects that might face overwhelming experiences more than others, were evaluated for the genetics and epigenetics of dopamine (*DAT1*) and serotonin (*SERT*) transporter genes, in addition to the expression of miR-132, miR-491, miR-16, and miR-135. **Results**: We found an increase in DNA methylation at one specific CpG site at *DAT1* 5’UTR in highly sensitive students reporting high levels of perceived stress when compared to those less sensitive and/or less stressed. Moreover, considering *DAT1* VNTR at 3’UTR, we observed that this effect was even more pronounced in university students having the 9/9 genotype when compared to those with the 9/10 genotype. These data are corroborated by the higher levels of miR-491, targeting *DAT1*, in highly sensitive subjects with high levels of perceived stress. *SERT* gene DNA methylation at one specific CpG site was reported to instead be higher in subjects reporting lower perceived stress when compared to more stressed subjects. Consistently, miR-135 expression, regulating *SERT*, was lower in subjects with higher perceived stress. **Conclusions**: We here suggest that the correlation of *DAT1* and *SERT* genetic and epigenetic data with the analysis of stress and sensitivity might be useful to suggest possible biomarkers to monitor mental health wellness in vulnerable subjects.

## 1. Introduction

Physiological and psychological challenges occur in life transition periods like late adolescence and young adulthood [1,2,3,4,5,6,7,8,9,10,11,12], during which coping with stress is necessary and natural. This is particularly true for university students since several environmental triggers, such as academic overload, insufficient organization of time, competitiveness, financial concerns, and familial pressures, produce stressful situations [13] that might alter mental health and well-being [14,15,16,17]. Indeed, several mental health disorders happen to start during this period [16,17,18,19,20].

The perception of stress varies among students [21] and the possible development of a mental health issue and/or a decrease in well-being might occur more likely in individuals with high sensitivity [22,23]. Highly sensitive individuals possess a keen awareness of their surroundings [24], become easily overwhelmed by an emotionally charged environment [25,26], and are therefore more susceptible to perceived stress [27]. From a biological perspective, a clear understanding of what induces a high sensitivity remains elusive [28]. Among others, genetic variations affecting serotonergic or dopaminergic systems have been reported to be relevant in subjects reporting environmental adversity stress contributing to high sensitivity [29,30,31]. It is worth mentioning the serotonin transporter-linked promoter region (5-HTTLPR) polymorphism, characterized by a variable number of tandem repeats (44-basepair insertion/deletion) in the serotonin transporter (*SERT*) gene promoter region, giving rise to a long (L) and short (S) allele. Numerous meta-analyses revealed a consistent relationship between a genetic susceptibility to anxiety and the polymorphism of the serotonin transporter gene [32,33,34,35]. In particular, the S allele is associated with anxiety-related traits [36] and, of relevance in the frame of this report, to neuroticism-related traits in a healthy population [37]. On the other hand, the L allele has been associated with increased anxiety in subjects during stressful life events [38]. Another relevant polymorphism located in the dopamine transporter (*DAT1*) gene at the 3′-untranslated region (UTR) has been associated with anxiety and stress [39,40]. It consists of a variable number of tandem repeats (VNTR) repeated between 3 and 13 times, with the 9- and 10-repeat alleles being the most frequent. Researchers have deeply investigated this VNTR in personality trait variation [41].

However, for these polymorphisms, as well as for many others, results are not consistent across studies, and most molecular genetic studies focusing on specific human traits generally account for less than 1% of individual variance [42]. This lack of reproducible findings is due to many factors, such as subjects’ different compositions (size, ethnicity, age, gender) and differences in the measures of behavioral outcomes. It should also be considered that it is very likely that complex traits, such as personality, might be polygenic, and it is of relevance to evaluate the possible effects of the combination of multiple gene loci [43,44]. Moreover, environmental factors could have a role in the development of a specific phenotype interacting with the genotype [45,46] or beside the genotype [47,48,49]. It is thus important to account for epigenetics, which involves all those marks that, without changing the DNA code, can alter the physical structure of the genome and, in turn, activate or inactivate genes [47].

Considering this background, the present study seeks to identify the risk factors that make university students more susceptible to developing mental health issues by examining if individual differences in sensory-processing sensitivity and perceived stress can be associated with differences in the genetic and epigenetics of *DAT1* and *SERT* genes. For assessing sensitivity, we referred to the phenotypical trait of sensory-processing sensitivity. This trait is different from other commonly assessed personality and temperament traits, and it helps predict how individuals respond to environmental stimuli [24,50]. For the molecular analysis, we obtained genomic DNA and exosomal circulating microRNAs (miRNAs) from non-stressful and easily accessible samples of saliva, already used to conduct genetic as well as epigenetic studies, specifically, DNA methylation [51] and miRNA expression [52,53] studies. Moreover, changes in DNA methylation in saliva have been found to correlate with those in the brain [54] and blood [55].

## 2. Materials and Methods

### 2.1. Participants

A total of 104 subjects (mean age of 20.04 ± 1.77 years; 19 males and 85 females) were recruited on a voluntary basis through a public announcement disseminated across various university courses. Exclusion criteria included a prior diagnosis of a psychiatric disorder, previous or current use of medication for a psychiatric disorder, and past or present use of drugs or psychotropic substances. Written informed consent was provided by participants and they also did not receive credit compensation for their participation. The sample’s characteristics are reported in Appendix A.

### 2.2. Procedure

Each participant completed an online survey (Qualtrics, Provo, UT, USA) that could be submitted just one time from an IP address. The survey included the Highly Sensitive Person (HSP) test and the Perceived Stress Scale (PSS) test as well as questions about demographic features (gender and age). Furthermore, to guarantee that participants were engaged throughout the survey process, a verification question was incorporated, prompting them to select a specific response. Questionnaires were completed independently rather than answering the operator to prevent interference in responding to the individual questions.

### 2.3. Behavioral Tests

#### 2.3.1. The 12-Item HSP Scale

The Highly Sensitive Person (HSP) test is an instrument developed to evaluate the sensory-processing sensitivity of stimuli coming from the environment [24]. In this study, the Italian version of the test was used with a self-report of 12 items [23,50] assessed on a 7-point Likert scale (from 1 = strongly disagree to 7 = strongly agree). Example items are “Are you easily overwhelmed by strong sensory input?”, “Do other people’s mood affect you?”, and “Are you deeply moved by arts or music?”. The instrument is composed of three psychometric subscales: ease of excitation (items 4, 6, 8, 9, 12), which is being easily overwhelmed by stimuli both internal and external; esthetic sensitivity (items 1, 3, 5, 10), which is capturing esthetic awareness; and low sensory threshold (items 2, 7, 11), which is unpleasant sensory arousal to external stimuli [56]. In line with the bifactor model, HSP has a total score obtained by taking the average across all 12 items, with higher scores indicating higher levels of sensitivity [57]. For the current sample, the measure’s internal consistency was good (Cronbach’s α = 0.81).

#### 2.3.2. Perceived Stress Scale (PSS-10)

The Perceived Stress Scale (PSS) is commonly used to measure stress levels [58]. In the current work, the Italian 10-item version of the scale was administered (PSS10) [59]. Respondents were asked to answer questions about their self-assessed psychological state during the last month using a five-point Likert scale (from never = 0 to very often = 4). An example of an item is, “How often have you been upset because of something that happened unexpectedly?”. The measure had good internal consistency in the present sample (Cronbach’s α = 0.79).

#### 2.3.3. Interaction between HSP and PSS

We correlated data on sensitivity and perceived stress with the molecular ones and defined new cut-offs based on HSP × PSS interactions. Students were identified as follows: a “low cumulative risk” group considering their low/medium sensitivity and their low/moderate perceived stress (HSP<4 and PSS<26, together with HSP<4.5 and PSS<13); a “medium cumulative risk” group considering a medium interval of both the parameters and the combination between low sensitivity and high perceived stress (HSP<4 and PSS>27) or between high sensitivity and low perceived stress (HSP>4.5 and PSS<13); and a “high cumulative risk” group considering both the highest values for the two parameters (HSP>4.5 and PSS>27), or the combination between the high score of one parameter and the medium score of the other one (HSP>4.5 and 14<PSS<26; 4<HSP<4.5 and PSS>27). This subjects’ stratification was considered for DNA methylation analysis at *DAT1* and *SERT* CpG sites and miRNAs expression.

### 2.4. Molecular Studies

#### 2.4.1. Salivary Samples Collection

Participants self-collected saliva samples by spitting about 2 mL into collection tubes. To minimize contamination risks, they were asked to not eat, drink (water permitted), take medications, use lip products, smoke, or brush their teeth for at least two hours prior to sample collection. The samples were then stored at −80 °C until further processing.

#### 2.4.2. DNA Extraction and Methylation Study

Genomic DNA was extracted by salting-out method [60] and assessed for quantity and quality using the NanoDrop 2000c UV-Vis Spectrophotometer (ThermoFisher Scientific, Waltham, MA, USA). DNA was bisulfite converted using EZ DNA Methylation-GoldTM Kit (Zymo Research, Orange, CA, USA), and each CpG site DNA methylation status was assessed by pyrosequencing [see [61] for details]. The reliability of the results is guaranteed by built-in controls for the bisulfite treatment, which removes the need for manual calculation of non-converted DNA levels and prevents false-positive methylation detection, including an internal control for the bisulfite treatment in the analysis to examine the DNA methylation percentage in the individual CpG site [62]. The ROUT method was used to exclude values recognized as outliers. The primers used and the genomic location of the sequence analyzed are reported in Table 1.

#### 2.4.3. Exosomal miRNAs Extraction and Real-Time PCR

Exosomal vesicles were isolated from saliva samples using Total Exosome Isolation Buffer (Invitrogen by Thermo Fisher Scientific, Waltham, MA, USA) according to the manufacturer’s recommended protocol. Saliva samples were centrifuged at 2000× *g* at 4 °C for 10 min, 500 µL of supernatant was collected and 250 µL of total exosome isolation buffer was added. After a 1 h incubation at 4 °C, exosomes were precipitated by centrifugation at 10,000× *g* for 1 h. The supernatant was discarded, and samples were subjected to additional centrifugation at 10,000× *g* for 5 min to remove the residual solution. The exosomal RNA was extracted using TRIzol Reagent (Life Technologies, Ambion, Austin, TX, USA) and the miRNAs were reverse transcribed with the miRCURY LNA RT Kit (Qiagen, Hilden, Germany) according to the manufacturer’s instructions.

The expression of miRNAs was measured by qRT-PCR on a DNA Engine Opticon 2 Continuous Fluorescence Detection System (MJ Research, Waltham, MA, USA) using SensiFAST™ SYBR Lo-ROX (Bioline—Meridian Bioscience, Tennessee, TN, USA) and specific primers (Qiagen, Hilden, Germany) (Table 2). We defined the relative amount of four miRNAs selected using the miRDB online database [63] for miRNA target prediction (two miRNAs targeting *DAT1*: miR-132 and miR-491; two miRNAs targeting *SERT*: miR-135 and miR-16-5p). Their expression was normalized using the ΔΔCt method (against the housekeeping miR-26a-5p and U6 snRNA) and then converted to the relative expression ratio (2^−ΔΔCt^).

#### 2.4.4. *SERT* 5-HTTLPR and *DAT1* VNTR Genotype

The following PCR primers were used for 5-HTTLPR genotyping: F: 5′-CGTTGCCGCTCTGAATGC-3′; R: 5′-TGGTAGGGTGCAAGGAGAATG-3′ [64]. Cycling conditions were as follows: 95 °C for 15 min, 94 °C for 30 s, 56 °C for 30 s, 72 °C for 30 s (45 cycles), and 72 °C for 10 min. A 2% agarose gel electrophoresis was used for the detection of PCR fragments visualized using Bio-Rad Gel Doc System: short allele, 340 bp; long allele, 384 bp. 

The following PCR primers were used to detect the 40 bp VNTR in the 3′-untranslated region of the *DAT1* gene: F: 5′-TGTGGTGTAGGGAACGGCCTGAG-3′; R: 5′-CTTCCTGGAGGTCACGGCTCAAGG-3′ [65]. PCR fragment lengths were 360 bp (7-repeat allele), 400 bp (8-repeat allele), 440 bp (9-repeat allele), and 480 bp (10-repeat allele). A schematic representation of the gene regions analyzed is reported in Figure 1.

#### 2.4.5. Statistical Analysis

The data are expressed as mean ± standard error of the mean (SEM). Non-parametric One-way ANOVA was used to test the difference between the groups in the individual CpG site for DNA methylation analysis and to evaluate the difference in miRNAs expression between the three groups of subjects identified by their PSS, HSP, and HSP × PSS scores. Differences in individual CpG site methylation were analyzed using Dunn’s test without multiple comparisons since none of the groups that are part of the study sample can be identified as a control group. The correlation analysis between DNA methylation levels in the individual CpG sites or between the two psychometric factors was performed with Spearman’s correlation coefficient. The Hardy–Weinberg equilibrium for genotype analysis was assessed using the Chi-square test. Considering the arbitrary division of the subjects based on questionnaire scores, we compared the groups one to one using the Kruskal–Wallis with Uncorrected Dunn’s test. The *p*-values < 0.05 were considered statistically significant.

## 3. Results

### 3.1. Sample Characteristics

Subjects were divided into three groups on the basis of their score on the PSS test (low (PSS<13, N = 18), medium (14<PSS<26, N = 55), high (27<PSS, N = 31)), the HSP test (low (HSP<4, N = 10), medium (4<HSP<4.5, N = 21), high (4.5<HSP, N = 73)) or considering the interaction between HSP and PSS (low cumulative risk (low HSP with low PSS, low HSP with medium PSS, and medium HSP with low PSS, N = 17); medium cumulative risk (medium HSP with medium PSS, low HSP with high PSS, and high HSP with low PSS, N = 21); high cumulative risk (high HSP with high PSS, medium HSP with high PSS, and high HSP with medium PSS, N = 66)) (see Appendix A).

### 3.2. DNA Methylation Studies

An increase in DNA methylation was observed at *DAT1* gene CpG 1 (Chr5: 1444717) in the high perceived stress group when compared to the moderate perceived stress group (medium = 7.40 ± 0.32; high = 8.34 ± 0.35, *p* = 0.0254; Kruskal–Wallis, Uncorrected Dunn’s test). At *DAT1* gene CpG 7 (Chr5: 1444685), higher DNA methylation levels were also observed in the moderate perceived stress group when compared to the high perceived stress group (medium = 11.79 ± 0.63; high = 9.92 ± 0.36, *p* = 0.0392) (Figure 2a). Instead, DNA methylation levels were lower in the high perceived stress group at *SERT* gene CpG 1 (Chr17: 30235932) when compared to the low perceived stress group (low = 3.24 ± 0.18; high = 2.85 ± 0.14, *p* = 0.0455) (Figure 2b).

Considering HSP, a significant increase in DNA methylation levels in students with high sensitivity compared to the medium sensitivity group was observed at *DAT1* CpG 5 (Chr5: 1444694) (medium = 8.82 ± 0.70; high = 11.45 ± 0.67, *p* = 0.0326) (Figure 3a). No differences were observed at *SERT* DNA methylation levels considering the HPS values (Figure 3b).

Based on the HSP × PSS interaction (Figure 4a), significantly higher levels of methylation at *DAT1* CpG 5 have been observed in the high cumulative risk compared to the medium cumulative risk group (medium = 8.30 ± 0.67; high = 11.54 ± 0.72, *p* = 0.0090) (Figure 4b). No differences among groups were observed for *SERT* DNA methylation levels (Figure 4c). The individual *p* values for each comparison are reported in Appendix A.

We correlated *DAT1* DNA methylation levels in individual CpG sites, as well as in the average (Ave) of the 7 CpG under study with “low”, “medium”, and “high cumulative risk” groups (Figure 5a, b, and c, respectively), based on the HSP × PSS interaction. Methylation levels at CpG 2 (Chr5: 1444714), CpG 3 (Chr5: 1444711), and CpG 6 (Chr5: 1444632) were significantly correlated within the high-risk group (Figure 5c). Notably, within the low-risk groups (Figure 5a), the CpG 6 was inversely correlated to CpG 5 although not significantly, probably for the lower number of subjects in this group. Of note, we previously reported that CpG 5 was inversely correlated with CpGs 3 and 6 [66,67].

### 3.3. Genetic Analysis

We finally analyzed the repetition frequency of the 40 bp-VNTR located into the 3’UTR of the *DAT1* gene and the *SERT* 5-HTTLPR polymorphism. The Chi-square test did not reveal a significant difference for the genotype as a function of the subjects’ HSP and PSS score classification (Appendix A), as well as for the interaction of the two parameters, HSP × PSS (Appendix A). When we looked at the allele distribution, we observed instead a massive presence of the 9-repeat allele within the medium PSS group with respect to the other two groups (Chi-square: 18.17, *p* = 0.0058) (Appendix A). No difference between the groups was observed when considering the 7-repeat, 8-repeat, and 10-repeat alleles.

We then considered correlating genotype distribution as a function of DNA methylation levels in the gene regions under study. In the *DAT1* 5’UTR considered for DNA methylation analysis, we observed higher DNA methylation levels at CpG 5 in the subjects with the 9/9 genotype (13.13 ± 1.58) compared to the 9/10 genotype (8.91 ± 0.40, *p* = 0.0332) (Figure 6a). The same analysis performed for the 5HTTLPR genotype as a function of *SERT* promoter region DNA methylation did not show any difference between the groups (Figure 6b).

### 3.4. miRNAs Analysis

We did not observe any difference in miR-132, miR-491, and miR-16-5p expression levels between all the subjects under study classified based on PSS and/or HSP scores (Figure 7a,b,d–h). Instead, we report a significant reduction in the expression of miR-135, targeting *SERT*, in subjects with high PSS with respect to the medium PSS group (medium: 1.67 ± 0.25; high: 1.12 ± 0.37, *p* = 0.0051, Kruskal–Wallis, Uncorrected Dunn’s test) (Figure 7c). When we considered the interaction of HSP × PSS parameters, an increase in the expression of miR-491, targeting *DAT1*, was observed in subjects with high scores (11.36 ± 3.33) when compared to those with medium scores (2.62 ± 1.15, *p* = 0.0196) (Figure 7j), while no differences between the groups under study were reported in the expression of the other miRNAs (Figure 7i,k,l). The individual *p* values for all the comparisons are reported in Appendix A. We then correlated subjects’ miRNA expression considering the individual low, medium, and high PSS, HSP, and HSP × PSS scores. For both the PSS and HSP parameters, we observed a strong direct correlation between miR-135 expression and both miR-132 and miR-491, both of which are involved in *DAT1* gene regulation. miR-135 showed a strong direct correlation with miR-132 considering the PSS (Spearman’s R = 0.438, *p* < 0.001) and the HSP (Spearman’s R = 0.438, *p* < 0.001), and the same correlation, also if weaker, was observed with miR-491 considering the PSS (Spearman’s R = 0.214, *p* = 0.0491) and the HSP parameters (Spearman’s R = 0.263, *p* = 0.0371) (Appendix A). Also considering the HSP × PSS interaction, again, we observed a direct relationship between miR-135 and miR-132 expression (Spearman’s R = 0.388, *p* < 0.001) (Appendix A).

## 4. Discussion

The first main result of our study is that *DAT1* gene regulation is affected in highly sensitive individuals, and this effect is emphasized in those reporting higher levels of stress. We report an increase in DNA methylation at one specific CpG site (number 5) at gene 5′UTR in highly sensitive individuals with higher PSS scores when compared to those less stressed. DNA methylation of the human *DAT1* gene has been extensively investigated since 80% of its promoter is formed by GCs sequences and it has multiple CpG islands [68,69], and alterations in the epigenetic mark have been reported in different phenotypes, such as ADHD [70], alcohol dependence [71], mild Internet use [66], and nicotine dependence [72], showing at specific CpG sites both reductions as well as increases in methylation levels [72,73,74]. In particular, the transition from anti-correlation in low-risk subjects towards direct correlation in high-risk subjects was similarly found in our previous studies on internet addiction [66,67]. Of note, in the 5′-UTR analyzed, CpG site 5 together with CpG site 6 represents a CGCG-core motif, a stress response motif regulated by transcription factors members of calmodulin-binding transcription activators (CAMTAs) [75,76]. The higher levels of miR-491 in highly sensitive individuals further corroborate our data on *DAT1* DNA methylation, both assuming the downregulation of the gene. Of note, miR-491 has already been reported to be involved in mental health [77].

Thus, both epigenetic mechanisms targeting *DAT1* might be relevant as potential biomarkers to detect and confirm in highly sensitive subjects the presence of high perceived stress.

We also observed differences in *DAT1* DNA methylation among the 3′-UTR VNTR genotypes. DNA methylation at CpG 5 is significantly higher in subjects with the 9/9 genotype, independently from the level of risk, compared to those with the 9/10 genotype. Moreover, HSP subjects with high levels of perceived stress carrying the 9/9 genotype have higher levels of *DAT1* DNA methylation and higher expression of miR-491. Different VNTR genotypes affect the level of expression of *DAT1* [78] and, consistent with our report, higher *DAT1* expression was found in the presence of the 10-repeat allele [79]. Our data assume lower DAT availability, and this has been observed at the preclinical level to be evoked by early life stress [80] and at the clinical level in major depression [81,82,83,84].

Overall, our data on *DAT1* transcriptional regulation could imply higher dopamine levels that might account for a mental health issue, since hyperdopaminergic activity characterizes several neuropsychiatric conditions.

We also support the possible role of *SERT* regulation in psychopathology [85,86,87]. In fact, *SERT* DNA methylation was reduced in subjects with higher levels of stress. Again, others already reported alterations in the epigenetic mark in individuals exposed to environmental stressful conditions [88], and in panic disorder patients [89], as well as in major depression associated with childhood adversities [90]. Moreover, we here report a potential role for miR-135, which might also be responsible for an increased *SERT* expression in subjects with higher stress levels. Of relevance, a previous study already observed the relevant role of miR-135 in chronic stress conditions associated with mood disorders [91]. Instead, we did not find any association of the 5-HTTLPR polymorphism with the different behavioral outcomes, in contrast with previous findings suggesting it is important in response to stress in female adolescents [92].

In conclusion, our data may be useful to suggest biomarkers that can bring attention to possible alarm bells for an alert condition. We hypothesize that integrating molecular data, genetics, and epigenetics with behavioral outcomes would help to identify subjects at greatest risk of developing mental health issues and thus provide possible resources for adaptive coping strategies to avoid maladaptive behaviors. In this context, the reversibility of epigenetic modifications is of great help as a possible preventive strategy through the manipulation of common environmental triggers such as diet and/or exercise. Of relevance, changes in environmental sensitivity are not only likely to predict maladjustment but can also be associated with better well-being when the environment is positive [93].

There are some limitations in our work that should be taken into consideration, like the low number of male participants. Future studies should investigate the feasibility of assessing sex differences in highly sensitive individuals. Moreover, in our analysis, we divided the population into different subgroups based on the PSS and HSP scores, and this enabled us to further stratify the analysis considering other sociodemographic variables.

## Figures and Tables

**Figure 1 biomedicines-12-02149-f001:**
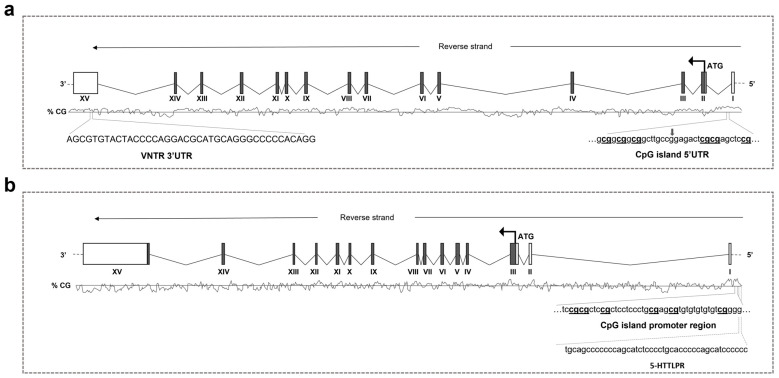
Schematic representation of the human *DAT1* (**a**) and *SERT* (**b**) genes. Boxes represent the exons, with the translated part filled in dark gray, ATG is also indicated. In the lower part of each panel, the CpG islands are reported, which are highlighted as the CpG sites under study as well as *DAT1* VNTR and *SERT* HTTLPR.

**Figure 2 biomedicines-12-02149-f002:**
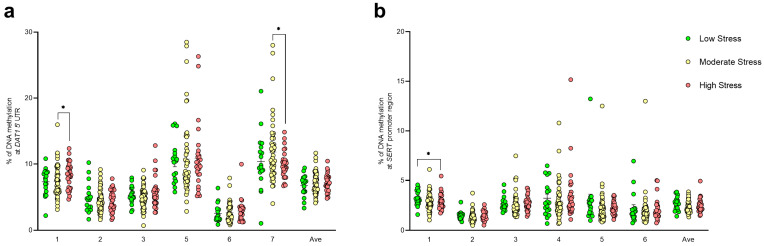
DNA methylation levels at *DAT1* (**a**) and *SERT* (**b**) CpG islands in saliva samples of young adults with PSS<13, 14<PSS<26, and PSS>27, represented as scattered dot plots (mean ± SEM, of each group) for the individual CpG sites and the average (Ave) of the CpG sites included in the study. Significant differences are indicated (* *p* < 0.05).

**Figure 3 biomedicines-12-02149-f003:**
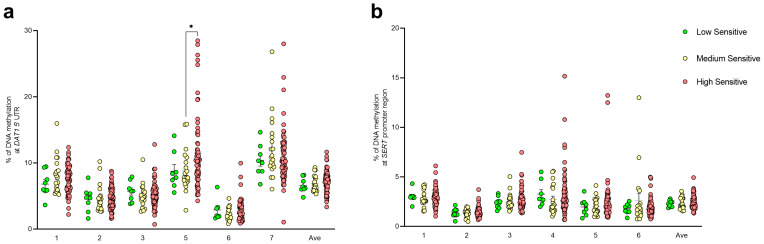
DNA methylation levels at *DAT1* (**a**) and *SERT* (**b**) CpG islands in saliva samples of young adults with HSP<4, 4<HSP<4.5, and HSP>4.5, represented as scattered dot plots (mean ± SEM, of each group) for the individual CpG sites and the average (Ave) of the CpG sites included in the study. Significant differences are indicated (* *p* < 0.05).

**Figure 4 biomedicines-12-02149-f004:**
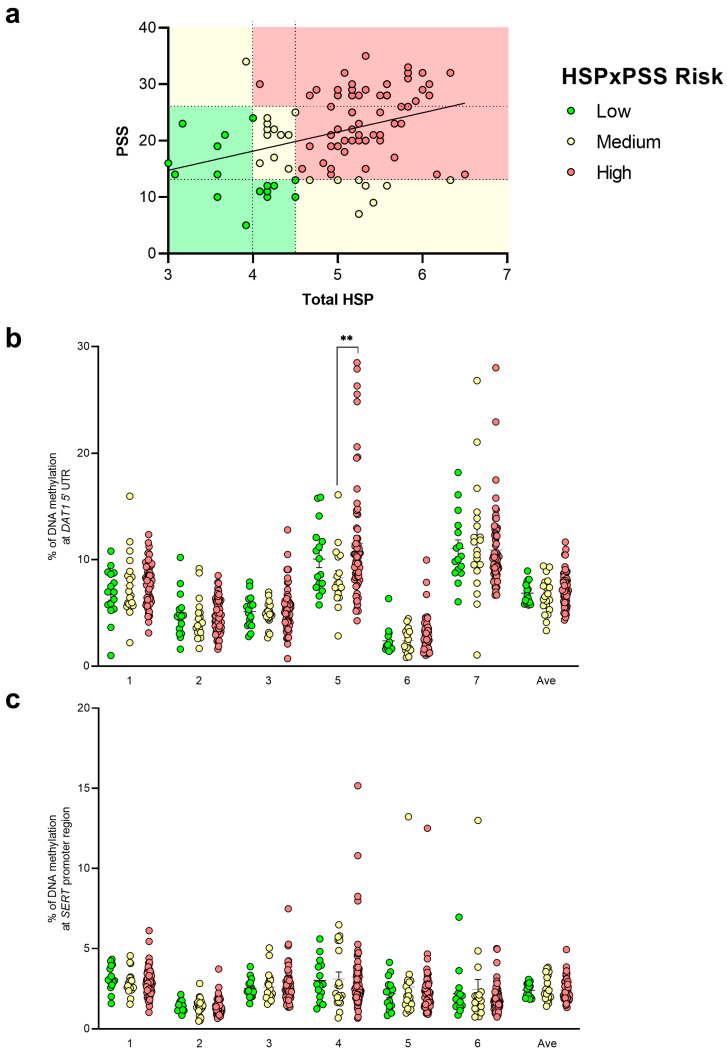
Subjects’ distribution considering the correlation between the PSS (Y axis) and the total HSP scores (X axis), determining 3 groups with a “low” (green circles), “medium” (yellow circles), and “high” cumulative risk (red circles) based on the interaction of the two parameters (**a**). DNA methylation levels at *DAT1* (**b**) and *SERT* (**c**) genes in saliva samples of young adults represented as scattered dot plots (mean ± SEM, of each group) for the individual CpG sites and the average (Ave) of the CpG sites included in the study. Significant differences are indicated (** *p* < 0.01).

**Figure 5 biomedicines-12-02149-f005:**
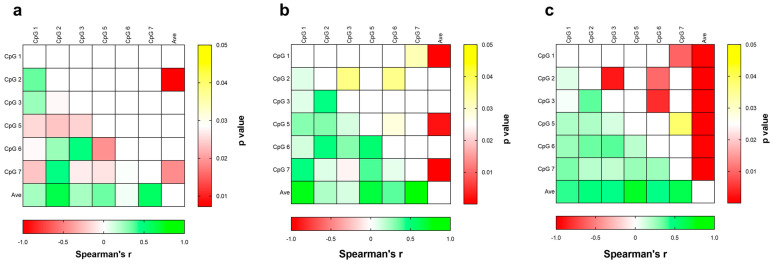
Heat maps representing the correlation analysis between DNA methylation levels at each CpG site of *DAT1* gene. Cells filled from green to red gradient (lower part) represent Spearman’s r; cells filled from yellow to red gradient (upper part) represent *p* values (empty cells represent *p* values greater than 0.05). Subjects under study are divided considering the correlation between the PSS and the total HSP scores, identifying a low (**a**), medium (**b**), and high (**c**) cumulative risk.

**Figure 6 biomedicines-12-02149-f006:**
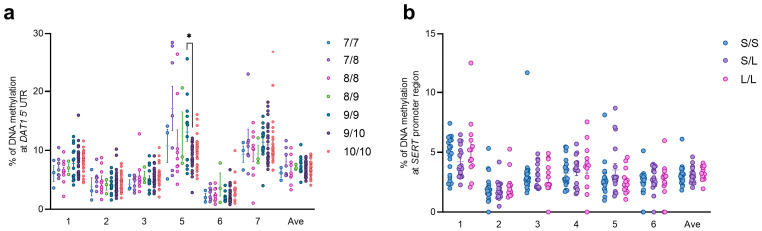
DNA methylation levels at *DAT1* (**a**) and *SERT* (**b**) CpG islands in saliva samples of young adults divided for their genotype at *DAT1* VNTR (**a**) and *SERT* 5-HTTLPR (**b**). Subjects are represented as scattered dot plots (mean ± SEM, of each group) for the individual CpG sites and the average (Ave) of the CpG sites included in the study. Significant differences are indicated (* *p* < 0.05).

**Figure 7 biomedicines-12-02149-f007:**
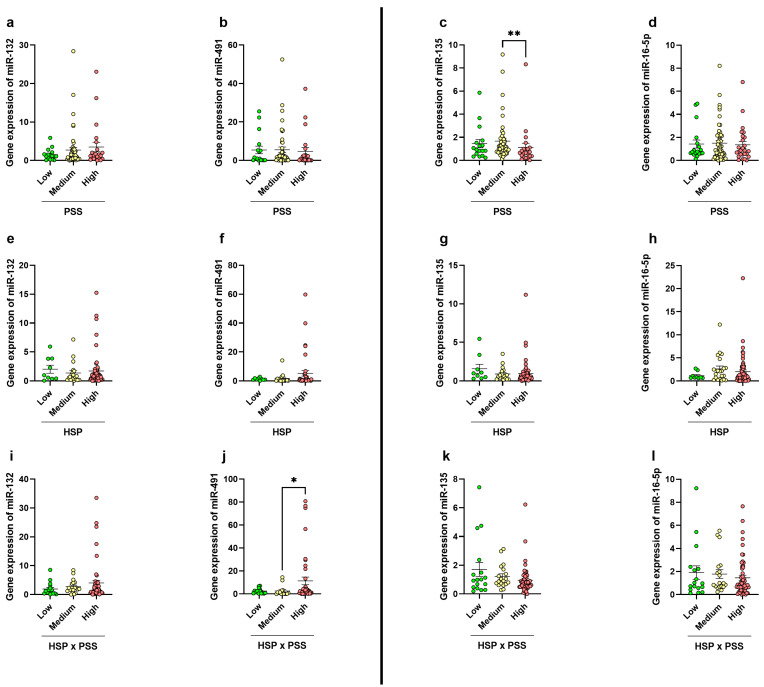
Expression levels of miR-132 (**a**,**e**,**i**), miR-491 (**b**,**f**,**j**), miR-135 (**c**,**g**,**k**), and miR-16-5p (**d**,**h**,**l**) in subjects reporting low, medium, or high score of PSS, HSP, and HSP × PSS. Significant differences are indicated (* *p* < 0.05; ** *p* < 0.01).

**Table 1 biomedicines-12-02149-t001:** Sequences considered for DNA methylation analysis. GeneGlobe Identification numbers refer to the assay designed for the analysis. Detailed genomic locations are reported.

Gene	GeneGlobe ID	Sequence Analysed	CpG Sites	Genomic Location
*DAT1*	PM00022064	GCGGCGGCGGCTTGCCRGAGACTCGCGAGCTCCGC	6	Chr5, bp 1444718-1444679
*SERT*	PM00065625	CCCCGACACACACACACGCTCGCAGGGAGGAGCGGAGCGCGGA	6	Chr17, bp 30235935-30235893

**Table 2 biomedicines-12-02149-t002:** Sequences considered for miRNA expression analysis. GeneGlobe Identification numbers refer to the assay designed for the analysis. miRBase accession codes and mature miRNA sequences are reported.

miRNA	GeneGlobe ID	miRbase Accession	Mature miRNA Sequence
hsa-miR-26a-5p	YP00206023	MIMAT0000082	UUCAAGUAAUCCAGGAUAGGCU
U6 snRNA	YP02119464	U6snRNA	
hsa-miR-132-3p	YP00206035	MIMAT0000426	UAACAGUCUACAGCCAUGGUCG
hsa-miR-491-5p	YP00204695	MIMAT0002807	AGUGGGGAACCCUUCCAUGAGG
hsa-miR-16-5p	YP00205702	MIMAT0000069	UAGCAGCACGUAAAUAUUGGCG
hsa-miR-135	YP00204762	MIMAT0000428	UAUGGCUUUUUAUUCCUAUGUGA

## Data Availability

Data will be made available on request.

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
