# Peer review of "Dopamine and Serotonin Transporter Genes Regulation in Highly Sensitive Individuals during Stressful Conditions: A Focus on Genetics and Epigenetics"

_biomedicines, 2024, doi:10.3390/biomedicines12092149_

Round 1
Reviewer 1 Report
Comments and Suggestions for Authors
The manuscript entitled “Dopamine and Serotonin Transporter Genes Regulation in Highly Sensitive Individuals Under Stressful Conditions: Focus on Genetics and Epigenetics” examines the differences in DNA methylation level in 7 CpG sites in promoter regions of the 5-HTT and DAT1 genes among 104 mentally healthy young adults with different person’s sensitivity and stress level. The authors made an attempt to find the relations between the genetic variants, DNA methylation level and microRNA expression level and stress sensitivity using various molecular-genetic approaches. A statistical analysis is appropriate and the manuscript represents a well-written paper with a significant number of figures that help to understand the findings. Although a sample size is rather small, the authors indicated this issue in limitations.
However, several issues need to be clarified.
1. The data on the number of examined individuals, age, values of statistical tests, designations of analyzed microRNAs, psychological tests used for the study are absent in the Abstract.
2. I suggest to add some data on meta-analysis studies of 5-HTTLPR and DAT1 VNTR polymorphisms in the Introduction, since there are a plethora of studies examining their effect on anxiety-related traits. In addition, it is required to clarify in the Introduction why the authors have selected certain microRNAs for the study.
I would suggest to accept after minor revision, providing that the authors addressed all the comments.
Author Response
Comment 1: The data on the number of examined individuals, age, values of statistical tests, designations of analyzed microRNAs, psychological tests used for the study are absent in the Abstract
Response 1: We appreciate the reviewer's suggestions, and we modified the abstract (lines 4-7) to include the number of participants examined as well as the microRNAs list. To avoid repetitions with the Materials and Methods paragraph, which contains all of the above data, we did not report additional details in the abstract.
Comment 2: I suggest to add some data on meta-analysis studies of 5-HTTLPR and DAT1 VNTR polymorphisms in the Introduction, since there are a plethora of studies examining their effect on anxiety-related traits. In addition, it is required to clarify in the Introduction why the authors have selected certain microRNAs for the study.
Response 2:
We thank the reviewer for the comment and in the Introduction, we have now included additional studies concerning DAT1 VNTR and 5-HTTLPR polymorphisms linked to anxiety-related traits (ref 32-35). The approach by which we picked these microRNAs to investigate is detailed in the paragraph “2.4.3. Exosomal miRNAs extraction and Real‐Time PCR”: We defined the relative amount of four miRNAs selected using miRDB online database for miRNA target prediction (two miRNAs targeting DAT1: miR-132 and miR-491; two miRNAs targeting SERT: miR-135 and miR-16-5p)
Reviewer 2 Report
Comments and Suggestions for Authors
In this study titled ‘Dopamine and serotonin transporter genes regulation in highly sensitive individuals under stressful conditions: focus on genetics and epigenetics’ by Bellia et al., the authors focused on dopamine transporter DAT1 and serotonin transporter gene SERT in young adults. They recruited a sample of 104 university student volunteers. They compared the genetic and epigenetic features of these two genes in highly sensitive students versus less sensitive ones. They report differences in DAT1 5’UTR DNA methylation and 3’UTR VNTR (variable number of tandem repeats), and in the levels of miR-491 (targeting DAT1) and miR-135 (targeting SERT). Based on these observations, the authors suggest that DAT1 and SERT might serve as biomarkers to monitor mental health in vulnerable populations.
Overall, the idea of the paper is interesting. The sampling here is good, as the authors managed to recruit more than 100 volunteers of very similar ages (20.04 +- 1.77). The major question I have is that in addition to the genetic / epigenetic traits that reported here, it seems that the authors didn’t check the gene expression levels of DAT1 and SERT (the mRNA levels which can be determined by RT-qPCR). This is straightforward but very important. Given the differences observed in several layers of epigenetic regulations, the readers would anticipate differences in the gene expression levels of DAT1 and SERT between the two groups.
I would recommend acceptance of this manuscript once the authors have this result.
Comments on the Quality of English LanguageMinor editing of English language required.
Author Response
Comment 1: The major question I have is that in addition to the genetic / epigenetic traits that reported here, it seems that the authors didn’t check the gene expression levels of DAT1 and SERT (the mRNA levels which can be determined by RT-qPCR). This is straightforward but very important. Given the differences observed in several layers of epigenetic regulations, the readers would anticipate differences in the gene expression levels of DAT1 and SERT between the two groups.
Response 1: We appreciate the reviewer's feedback. We certainly agree that the differences in DNA methylation levels observed can be validated by gene expression analysis. Unfortunately, although we know that this analysis would add value to the current work, we were unable to carry out gene expression analysis. From saliva samples we have been able to obtain genomic DNA and exosomes of high quality but not high-quality tRNA to run this analysis. Furthermore, our objective was also to suggest biomarkers using samples undergoing to procedure where the molecular material is more stable as possible, and this is the case for DNA and exosomes.
Round 2
Reviewer 2 Report
Comments and Suggestions for Authors
I have no further comments.